# Reconstruct Before Summarize: An Efficient Two-Step Framework for Condensing and Summarizing Meeting Transcripts

**Haochen Tan[1], Han Wu[1], Wei Shao[1], Xinyun Zhang[2],**

**Mingjie Zhan[3], Zhaohui Hou[3], Ding Liang[3], Linqi Song[1]***

[1]Department of Computer Science, City University of Hong Kong
[2] Department of Computer Science and Engineering, The Chinese University of Hong Kong
[3]SenseTime Research
`haochetan2-c@my.cityu.edu.hk, linqi.song@cityu.edu.hk`

## Abstract

Meetings typically involve multiple participants and lengthy conversations, resulting in redundant and trivial content. To overcome these challenges, we propose a two-step framework, Reconstruct before Summarize (RbS), for effective and efficient meeting summarization. RbS first leverages a self-supervised paradigm to annotate essential contents by reconstructing the meeting transcripts. Secondly, we propose a relative positional bucketing (RPB) algorithm to equip (conventional) summarization models to generate the summary. Despite the additional reconstruction process, our proposed RPB significantly compressed the input, leading to faster processing and reduced memory consumption compared to traditional summarization methods. We validate the effectiveness and efficiency of our method through extensive evaluations and analysis. On two meeting summarization datasets, AMI and ICSI, our approach outperforms previous state-of-the-art approaches without relying on large-scale pre-training or expert-grade annotating tools.

## 1 Introduction

Although numerous achievements have been made in the well-structured text abstractive summarization (Zhang et al., 2020a; Liu* et al., 2018; Lewis et al., 2020), the research on meeting summarization is still stretched in limit. There are some outstanding challenges in this field, including 1) much noise brought from automated speech recognition models; 2) lengthy meeting transcripts consisting of casual conversations, content redundancy, and diverse topics; 3) scattered salient information in such noisy and lengthy context, posing difficulties for models to effectively capture pertinent details.

To this end, previous works adapt the language model to long inputs through techniques such as long-sequence processing (Beltagy et al., 2020; Tay et al., 2020; Zhong et al., 2022) and hierarchical

learning (Zhu et al., 2020; Rohde et al., 2021), or tailor the input to an acceptable length through sentence compression (Shang et al., 2018a) and coarse-to-fine generation (Zhang et al., 2022). However, these approaches do not specifically target the critical information in meeting transcripts. Feng et al. (2021b) utilizes the token-wise loss as a criterion to annotate contents with DialoGPT (Zhang et al., 2020b), suffering from labeling unpredictable contents as critical information. Besides, the commonly used pre-processing procedure that extends models' positional embedding by copying, and truncating the lengthy input compromises the positional relationships learned during pre-training, and results in a loss of important information due to the brutal truncation. Consequently, a natural question is - How can we precisely capture the salient contents from noisy and lengthy meeting transcripts, and summarize them with conventional language models?

Our observation is that meetings are characterized by extensive communication and interaction, with specific texts often containing pivotal content that drives these interactions. Based on this understanding, we propose a two-step meeting summarization framework, Reconstrcut before Summarize(RbS), to address the challenge of scattered information in meetings. RbS adopts a reconstructor to reconstruct the responses in the meeting, it also synchronically traces out which texts in the meeting drove the responses and marks them as essential contents. Therefore, salient information is captured and annotated as anchor tokens in RbS. To preserve the anchors but compress the lengthy and noisy input, we propose the relative positional bucketing (RPB), a dynamic embedding compression algorithm inspired by relative positional encoding (RPE) (Shaw et al., 2018; Huang et al., 2019). Our RPB-integrated summarizer can preserve the anchors and compress the less important contents according to their relative position to the

---

*Corresponding author.

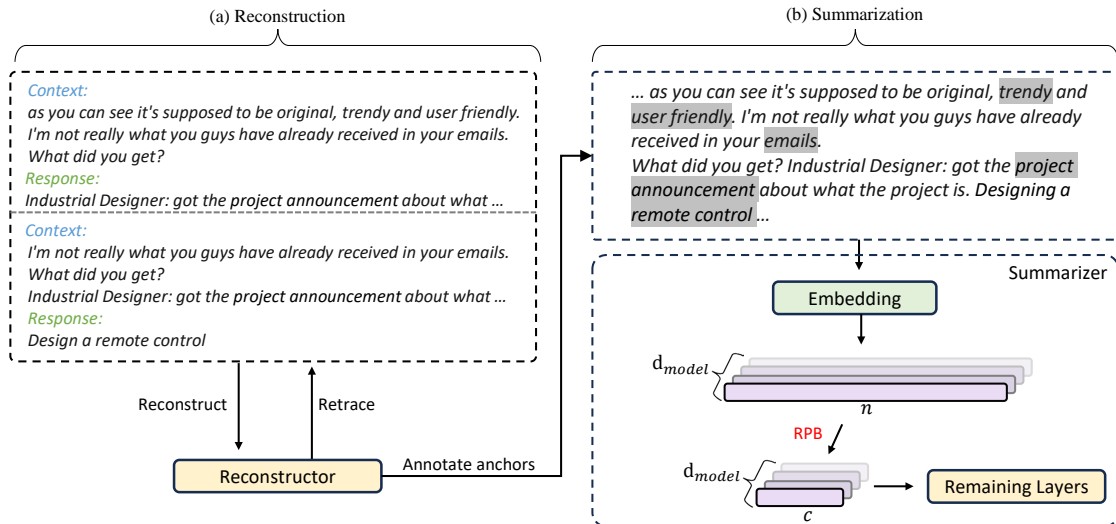

Figure 1: By computing the scaled attention, the reconstructor identifies the contribution of each token in the contexts toward recovering the response. Tokens that make significant contributions are marked as anchors(labeled grey in the figure). Following this, the summarizer embeds the annotated texts. RPB then compresses the embedding from $n \times d_{model}$ to $c \times d_{model}$ based on the anchors, where $n$ is the length of the input(normally 5k - 20k), $c$ is a constant(1024 by default) and $n \gg c$.

anchors. This allows the summarizer to generate a concise and informative summary of the meeting transcripts.

Although RbS introduces an additional importance assessment step, the introduction of RPB greatly compresses the length of the original text, making RbS even faster and memory-efficient than the traditional one-step approach. The experimental results on AMI (Mccowan et al., 2005) and ICSI (Janin et al., 2003) show that RbS outperforms previous state-of-the-art approaches and surpasses a strong baseline pre-trained with large-scale dialogue corpus and tasks. Extensive experiments and analyses are conducted to verify the effectiveness and efficiency of each process of our approach.

To sum up, our contributions are as follows: (1) We propose the RbS, an efficient and effective framework for long-text meeting transcripts summarization; (2) Without external annotating tools or large-scale pre-training corpus and tasks, our method can efficiently generate meeting minutes with conventional language models (PLMs); (3) Extensive experiments demonstrate the effectiveness of our framework.

## 2  Methods

The main architecture is shown in Figure 1. Our framework comprises two components: the reconstructor and the summarizer. The reconstructor is responsible for reconstructing meeting transcripts

and identifying the context that drives the interaction. Meanwhile, before generating the summary, the summarizer compresses the lengthy input and preserves critical content.

### 2.1  Reconstruction and Retracing

To capture the essential information, we propose retracing the contexts that drive interactions with a reconstructor. We split the meeting transcripts into context-response pairs. By reconstructing the response based on the context and tracking the contributing contexts, we can effectively capture the important content of the transcript.

**Reconstruction**  The architecture in Figure 1 illustrates the process. To recover each response, we use a window of size $w$ to limit the input history, with $w$ set to 3 in Figure 1. We assume that a meeting transcript contains $m$ sentences and create a sub-dataset consisting of $m$ pairs, i.e.,$\{S_{[max(0,i-w):i-1]}, S_i\}$, where $i \in [2:m]$. To prompt the language model to predict the end of the meeting, we add a special token [EOM] at the end of the transcript. Finally, a reconstructor recover the response from $S_w$ to [EOM] as closely as possible using the input $S_{[max(0,i-w):i-1]}$. The reconstruction is conducted via the forward pass of language model with teacher forcing (Williams and Zipser, 1989; Lamb et al., 2016).

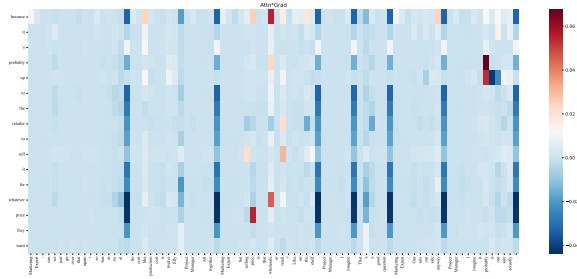

Figure 2: For context-response generation, RbS utilizes the scaled-attention to retrace how much each token in context contributes to the recovery of the response.

**Retracing**   As shown in Figure 2, during the reconstruction, RbS synchronically retrace the contribution of each token of the context to the recovery of the response, from the perspective of attention weights (Bahdanau et al., 2014; Kim et al., 2017; Vaswani et al., 2017) and gradients. Recap the procedure of the attention mechanism, the cross attention (Vaswani et al., 2017) is formulated as:

$$Attention(Q, K, V) = softmax(\frac{QK^T}{\sqrt{d_k}})V, \quad (1)$$

Where $Q$ is the representation of the response to be generated in the decoder, $K$, and $V$ are the memories and values that come from encoded contexts. Inspired by the works that adopt the attention and gradients to retrace which part of the input drives the model to make predictions (Jain et al., 2020; Kindermans et al., 2016; Atanasova et al., 2020; Sundararajan et al., 2017), we extract the importance scoring with scaled attention (Serrano and Smith, 2019) $a\nabla a$ from the last cross-attention layer to determine the contribution of each token in contexts $S_{[max(0,i-w):i-1]}$ to the restoration of the response $S_i$. The scaled attention $a\nabla a$ is denoted as the attention scores $a_i$ scaled by its corresponding gradient $\nabla a_i = \frac{\partial \hat{y}}{\partial a_i}$, where $\hat{y}$ is the model's prediction.

**Scores Aggregation**   We utilize a context window of size $w$ to reconstruct responses, which leads to reasonable reconstruction. However, this also results in each sentence being treated as context $w$ times to recover responses, which poses a challenge in combining importance-related scores during tracebacks and scoring. To address this issue, we carefully propose the hypothesis that each token can be considered to be scored according to $w$ different criteria after it is rated by $w$ different responses. Therefore, two different strategies are investigated in this paper, which we refer to aver-

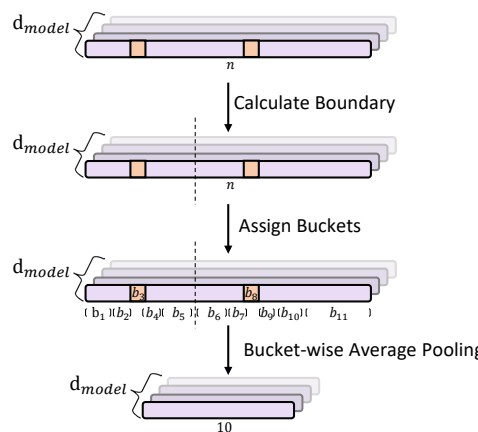

Figure 3: An illustration of compressing a $d_{model} \times n$ sequence embedding with two annotated anchors to $d_{model} \times 10$ using RPB, where $b$ is the bucket number. Two orange blocks are the anchors.

aging and multi-view voting. Averaging involves taking an average of the $w$ ratings for each token during the reconstruction process. We then select the top-$k$ tokens with the highest average rating as the salient information, which we refer to as **anchor** tokens in RbS. This approach leverages the average score to express the overall contribution of each token. Multi-view voting involves selecting the top-$\frac{k}{m}$ tokens with the highest score in each criterion after the reconstruction is completed. This approach considers multiple perspectives for evaluating contexts, selecting the contexts that contribute most prominently under each perspective as anchors.

## 2.2   Summarization

With the obtained anchors, our hypothesis is that tokens in meeting texts are more relevant to salient information when they are closer to the anchors, and conversely, tokens that are farther away from the anchors are less relevant to the important content. Therefore, we propose relative positional bucketing(RPB), which compresses the original input as losslessly as possible by preserving the anchors and dynamically compressing the less-important contexts around the anchors.

**Relative Positional Bucketing**   RbS employ an conventional language model that accept $c$ tokens input as the summarizer, consider a sequence $\{t_0, t_1, \cdots, t_m\}$ annotated with $n$ anchors $\{a_0, a_1, \cdots, a_n\}$, where $m \gg c \geq n$, and positions of all anchors are $\{i_0, i_1, \cdots, i_n\}$. The summarizer first extract the embeddings

$\{e_0, e_1, \cdots, e_m\}$ of the sequence. Then it identified positions $\{i_0 + \frac{i_1 - i_0}{2}, i_1 + \frac{i_2 - i_1}{2}, \cdots, i_{n-1} + \frac{i_n - i_{n-1}}{2}\}$ in the middle of all adjacent anchors as boundaries. Each pair of adjacent boundaries will form a sub-sequence containing an anchor point. For each sub-sequence, the summarizer obtains the position code of each token relative to the anchor. Inspired by the T5 (Raffel et al., 2020) that translates relative position to a bucket number for memory-efficient and long-sequence-friendly attention, we compress the sequence by bucketing the embeddings $\{e_0, e_1, \cdots, e_m\}$ with $c$ buckets.

We assign larger buckets to the embeddings that have a large relative distance to the anchors, and smaller buckets to the embeddings that are close to the anchors. Embeddings that share the same bucket will be compressed by average pooling. The calculation of bucket assignment for each token refers to Algorithm 1. Finally, the summarizer processes the compressed embeddings $\{e'_0, e'_1, \cdots, e'_c\}$ and generate the summary. Figure 3 demonstrates an example that RPB compresses $d_{model} \times n$ embeddings containing two anchors to $d_{model} \times 10$ embeddings. Such a process forms a dynamic compression based on the importance of the contexts.

The difference between our proposed summarizer and the original BART is only the addition of the RPB module, which is inserted between the embedding layer and the first attention layer. The summarization is based on the compressed embedding instead of the original one, which greatly saves memory and computation time. Furthermore, it is noteworthy that the bucketing operation and batch average-pooling are parallel constant calculation[1] and scatter-reduce operation[2], respectively, making the process highly efficient.

## 3 Experiments

### 3.1 Setup

**Dataset & Preprocessing** RbS is evaluated on two meeting summarization datasets, AMI (Mccowan et al., 2005) and ICSI (Janin et al., 2003). AMI is a dataset of business project meeting scenarios that contains 137 transcripts. The average length of input and target length is 6,007 and 296, respectively. ICSI is aimed at academic discussion scenarios, where professors and other students have discussions with each other. The average length

---

[1] T5 implementation
[2] scatter-reduce

of input and target reaches 13,317 and 488.5, respectively, while only 59 meeting transcripts are included. Following the preprocessing pipeline proposed in Shang et al. (2018a), we split the data into training/development/testing sets with the list provided in (Shang et al., 2018b): 97/20/20 for AMI and 42/11/6 for ICSI. Besides the meeting minutes, decisions, actions (progress in the ICSI), and problems encountered in the meeting are included in the golden summarization. Extra spaces and duplicate punctuation removal are adopted to clean the data further.

**Baseline & Metric** BART (Lewis et al., 2020) is selected as both the baseline and backbone. BART-CNN that finetuned on the CNN-Daily Mail (Hermann et al., 2015) is also evaluated. Sentence Gated (Goo and Chen, 2018) utilizes the dialogue acts to generate summaries. PGNet (See et al., 2017) is a traditional approach that summarizes the meeting with the pointer network. HMnet (Zhu et al., 2020) adopts cross-domain pre-training before summarizing with a hierarchical attention mechanism. HAT (Rohde et al., 2021) performs a hierarchical attention transformer-based architecture. DDAMS (Feng et al., 2021a) incorporates discourse information to learn the diverse relationship among utterances. Summ^N (Zhang et al., 2022) performs the split-then-summarize in multi-stage for lengthy input. Feng et al. (2021b) employs DialogGPT as the annotator to label the keywords and topics in the meeting transcripts. Besides, DialogLM (Zhong et al., 2022) pre-trained on large-scale dialogue-related corpus and tasks are also compared to show our efficiency. All approaches are evaluated with ROUGE (Lin, 2004), namely ROUGE-1, ROUGE-2, and ROUGE-L.

**Implementation Details** We use the released BART checkpoints in Huggingface's Transformers (Wolf et al., 2020) for RbS. Specifically, we initialize RbS with BART-large checkpoints, and the parameters in RbS-CNN are initialized by BART-large-CNN. During the response reconstruction, we used eight sentences as contexts. The reconstructor is trained for $2,300$ steps on the split AMI and $1,500$ steps on the split ICSI, with a learning rate of 5e-5 and a total batch size of 256. Once the reconstructor is enabled to recover the response, we perform one forward pass with the teacher-forcing to retrace the contribution of the contexts. During this process, $6.4\%$ of the tokens are annotated as

| Model | AMI | | | ICSI | | |
|---|---|---|---|---|---|---|
| | R-1 | R-2 | R-L | R-1 | R-2 | R-L |
| *Backbone* | | | | | | |
| BART-large (Lewis et al., 2020)($l = 3072$) | 49.99 | 16.95 | 47.79 | 43.70 | 9.77 | 41.34 |
| BART-large-CNN (Lewis et al., 2020)($l = 3072$) | 50.46 | 17.00 | 48.28 | 46.06 | 10.38 | 43.86 |
| *LSTM and RNN* | | | | | | |
| PGNet (See et al., 2017) | 42.60 | 14.01 | 22.62* | 35.89 | 6.92 | 15.67* |
| Sentence-Gated (Goo and Chen, 2018) | 49.29 | 19.31 | 24.82* | 39.37 | 9.57 | 17.17* |
| *Language Model as Annotator* | | | | | | |
| PGN (Feng et al., 2021b) | 50.91 | 17.75 | 24.59* | - | - | - |
| *Transformers* | | | | | | |
| HMNet (Zhu et al., 2020)($l = 8192$) | 52.36 | 18.63 | 24.00* | 45.97 | 10.14 | 18.54* |
| HAT-BART (Rohde et al., 2021)($l = 3072$) | 52.27 | 20.15 | 50.57 | 43.98 | 10.83 | 41.36 |
| DDAMS (Feng et al., 2021a)($l = 15000$) | 53.15 | **22.32** | 25.67* | 40.41 | 11.02 | 19.18* |
| Summ^N (Zhang et al., 2022) | 53.44 | 20.30 | 51.39 | 45.57 | 11.49 | 43.32 |
| *Large-Scale Dialogue-Specific Pre-train* | | | | | | |
| DialogLM (Zhong et al., 2022)($l = 5120$) | 54.49 | 20.03 | 51.92 | 49.25 | 12.31 | 46 .80 |
| *Ours* | | | | | | |
| RbS ($l = 1024$) | 54.06 | 21.02 | 52.07 | **50.28** | **13.24** | **47.15** |
| RbS-CNN ($l = 1024$) | **54.99** | 20.98 | **52.40** | 49.61 | 12.20 | 46.97 |

Table 1: The performance on AMI and ICSI. $l$ is the maximum number of input tokens for the corresponding model. * denotes the metrics are calculated without sentence split. RbS takes the BART-large as the backbone, while the backbone of RbS-CNN is BART-large-CNN.

anchors. The total bucket number is equal to the maximum acceptable input of the backbone, which is 1024 for BART. The quantity of buckets for each sub-sequence depends on the length ratio to the total length. For the summarizer, we set the learning rate as 3e-5 with a total batch size of 64. It is worth noting that RbS is trained solely on AMI and ICSI without any external data or tools. We do not introduce any pretraining from other domains.

## 3.2 Main Results

Table 1 presents the ROUGE scores on AMI and ICSI. Our framework outperforms baselines across all metrics. Notably, our model achieves a significant improvement on AMI (ROUGE-1 $49.99 \rightarrow 54.06$) and a more substantial gain on ICSI (ROUGE-1 $43.70 \rightarrow 50.28$), compared to the BART-large model. This is due to the longer context in ICSI, which exceeds 10K words on average, demonstrating the remarkable ability of our model to handle extreme-length inputs. When using BART-large-CNN as the backbone, we observe further improvements in the AMI dataset. Meanwhile, our RbS-CNN outperforms the previous state-of-the-art approach, Summ^N, by approximately 1.5 ROUGE-1 score on AMI, and 4 ROUGE-1 score on ICSI, without requiring the

large-scale dialogue-specific pre-training. Even when compared to DialogLM, which is pre-trained on large-scale dialogue-specific corpus and tasks and requires more time-consuming and computation resources, the advantages are still pronounced in ICSI datasets with longer input. The results demonstrate that RbS enhances conventional language models' ability to summarize lengthy meeting transcripts by focusing on salient information and compressing irrelevant content. Without extending the models' acceptable input length, RbS enables the conventional language model to summarize the meeting transcripts with less memory consumption.

## 4 Analysis

In this section, we conduct further analysis to show the effectiveness of the RbS. We aim to investigate the correlation between anchor tokens and salient information in the meeting transcript. Through extensive experiments, we will demonstrate the validity of our approach in capturing anchor tokens, and the significance of anchor tokens in conveying important information. Furthermore, we will analyze the impact of different methods for aggregating the importance scores. We will also provide justification for our bucketing algorithm, analysis

| Model | AMI | | |
|---|---|---|---|
| | R-1 | R-2 | R-L |
| RbS-CNN | **54.99** | 20.98 | **52.40** |
| Substitute - Random | | | |
| 25% | 52.47 | 19.12 | 50.33 |
| 50% | 52.84 | 19.74 | 52.14 |
| 75% | 51.03 | 18.81 | 48.76 |
| Substitute - High Frequency | | | |
| 25% | 53.60 | 20.32 | 51.33 |
| 50% | 53.38 | 20.15 | 51.15 |
| 75% | 52.84 | 20.70 | 50.75 |
| Deletion - Random | | | |
| 25% | 51.42 | 20.10 | 49.42 |
| 50% | 51.79 | 19.26 | 50.04 |
| 75% | 49.97 | 18.78 | 48.32 |
| Deletion - Sorted Fraction | | | |
| 0%-25% | 49.10 | 17.76 | 47.03 |
| 25%-50% | 50.97 | 17.28 | 48.89 |
| 50%-75% | 53.48 | 20.57 | 51.18 |
| 75% - 100 % | 53.19 | **21.30** | 51.20 |

Table 2: Ablation studies on the substitution and deletion of anchor tokens

of the computing complexity is also provided to prove the efficiency of the framework. Additionally, the potential for reusing the parameters of the reconstructor is explored in Appendix A.3.

## 4.1 Importance Scoring

In this section, we examine the impact of importance scoring on our framework. To demonstrate the criticality of the anchors selected by our reconstruction and retracing process, we conduct experiments in various settings: (1) We delete or substitute the selected anchors with different ratios and observe the resulting changes in performance. (2) We test the framework with different indicators of importance, including the attention weights, the gradient of the attention weights, random scoring, and token-wise loss similar to Feng et al. (2021b), which uses $r$ percentage of words with the highest reconstruction loss as keywords. (3) We extract and visualize the heatmap of our approach to see if the anchor words are precisely those we need. (4) We investigate the number of anchor tokens required to achieve acceptable performance for different scoring algorithms.

**Anchor Deletion and Substitution** For substitution, we take two measures. One is to replace the anchor tokens with other tokens randomly sampled from the meeting transcript, and the other

| Model | AMI | | |
|---|---|---|---|
| | R-1 | R-2 | R-L |
| Scaled Attention | **54.99** | **20.98** | **52.40** |
| Attention | 52.08 | 18.96 | 50.79 |
| Gradient | 53.41 | 20.35 | 50.80 |
| Token-wise Loss | 52.02 | 19.61 | 50.13 |
| Random | 50.84 | 18.64 | 48.88 |

Table 3: Ablation studies on different importance scoring approaches

is to replace anchors with high-frequency tokens. For anchor token deletion, there are also two different strategies. One is to delete the anchor tokens randomly, and the other is to sort the anchors in descending order of importance score and divide them into four fractions. Then, we remove one at a time to observe the performance change. The results in Table 2 demonstrate that both anchor token substitution and deletion negatively affect the performance. Specifically, when randomly substituting the anchor tokens with the others, a plunge of the ROUGE-1 scores could be observed ($54.99 \rightarrow 52.47$). Although the score improves slightly after replacing anchors with high-frequency tokens, the performance still falls far short of anchor tokens. This indicates that anchor tokens selected by our framework are informative and play irreplaceable roles. This phenomenon is even more evident in the random removal of the anchor tokens. Results on the performance of different percentages of anchor tokens also show that our framework produces strongly importance-correlated rankings.

**Attention, Gradient, and Token-wise Loss** We conduct an ablation study on different types of scoring indicators, namely the attention weights, gradients of the attention, and token-wise loss. Attention weights and their corresponding gradients are extracted from the last transformer layer of the model. As for the token-wise loss, different from our framework that treats the response as a query to rate the importance of the context, this approach scores the response directly according to the generation loss:

$$l = -\log(\frac{\exp(\hat{t}, t)}{\sum_{v=1}^{V} \exp(\hat{t}, v)}), \qquad (2)$$

where $\hat{t}$ is the generated token, $t$ is the ground truth, and $V$ is the vocabulary size. Similar to our setting, all the methods extract 6.4% tokens as anchors.

As shown in Table 3, scaled attention achieves the best performance on AMI. The performance of

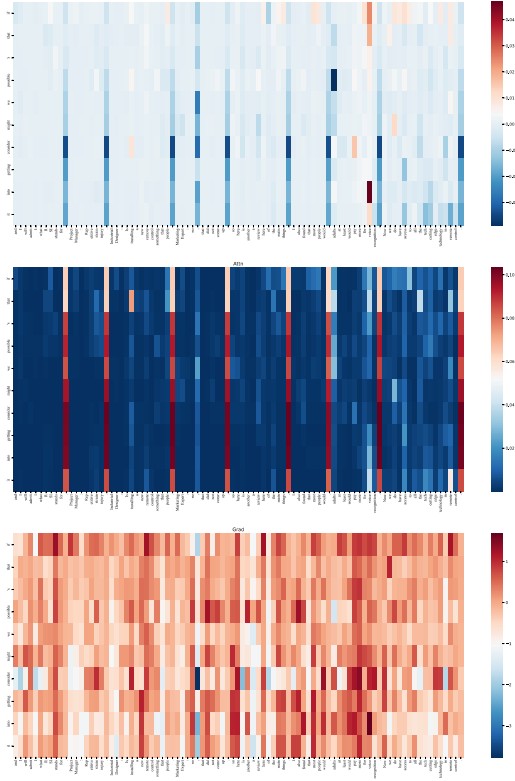

Figure 4: Visualization of the heatmap. From top to bottom are heatmaps of scaled attention, gradient, and attention weights, respectively.

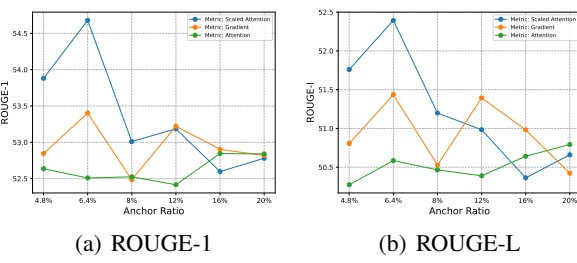

(a) ROUGE-1        (b) ROUGE-L

Figure 5: Trend of ROUGE score of different methods with increasing anchor ratio

the gradient is comparable with the scaled attention, while there are sharp decreases when switching the scaled attention to attention weights and token-wise loss. These results demonstrate that scaled attention weights are more importance-correlated than the others. This finding is consistent with Chrysostomou and Aletras (2022); Serrano and Smith (2019)

**Visualization** To validate the importance scoring approaches, we visualized the ratings of context in meeting transcripts. Figure 4 displays the heatmap for each scoring method using the response "if that's possible, we might consider getting into it," where "it" refers to voice recognition and cutting-

| Model | Aggregation | AMI | | |
|---|---|---|---|---|
| | | R-1 | R-2 | R-L |
| RbS | Vote | **54.06** | **21.01** | **52.07** |
| | Avg | 52.56 | 20.12 | 50.34 |
| RbS-CNN | Vote | **54.99** | **20.98** | **52.40** |
| | Avg | 53.88 | 20.61 | 51.77 |

Table 4: Ablation study on channel aggregation

edge technologies. This excerpt is from a meeting discussing the need to include voice recognition in remote controls. The middle of Figure 4 shows that most recovered tokens assign high attention scores to punctuation, indicating that attention weights do not accurately reflect context importance. The bottom part of the figure shows that while gradients can select essential content, they also assign high weights to irrelevant contents, making them an unsatisfactory indicator of importance. The top of Figure 4 shows that scaled attention weights accurately detect important content, assigning high scores to tokens such as "pay more for voice recognition" and "cutting-edge technology in remote control," while giving low scores to most other content, especially punctuation. This visualization provides an intuitive picture of our framework's ability to capture key points, further explaining its superior performance.

**Number of anchors** We conduct the ablation studies on how the number of anchors influences the framework's performance, as shown in Figure 5. We gradually increased the number of anchor points and observed the change in the ROUGE score. Surprisingly, we found that the total number of anchors did not need to be very high; in fact, increasing the number of anchor tokens resulted in performance degradation. We attribute this phenomenon to the fact that the total number of buckets for the BART model is limited to 1024. The more anchor tokens there are, the fewer buckets the other tokens can share, leading to over-compressed context and performance degradation. We also observed that our method achieved strong performance with fewer top-ranked tokens, while the other two methods required more anchor tokens to achieve acceptable performance. This indicates that our approach effectively captures salient information.

| Truncation | AMI | | |
|---|---|---|---|
| | R-1 | R-2 | R-L |
| Bucketing | **54.99** | **20.98** | **52.40** |
| Right | 50.46 | 17.00 | 48.28 |
| Middle | 51.18 | 19.40 | 50.78 |
| Left | 50.59 | 18.32 | 48.34 |
| Random | 48.79 | 18.03 | 47.13 |
| Hard truncation | 51.90 | 18.00 | 49.93 |

Table 5: Ablation study on bucketing and truncation

### 4.2 Scores Aggregation

We conduct an ablation study on two proposed score aggregation methods: averaging and multi-view voting. Results in Table 4 show that multi-view voting outperforms averaging. We attribute this to the fact that averaging disrupts the multi-perspective rating mechanism. This result is consistent with our motivation that multi-view voting brings multiple horizons to the choice of anchor tokens. Therefore, we conclude that multi-view voting is necessary and beneficial for filtering anchor tokens.

### 4.3 Bucketing and Truncation

Our bucketing strategy can be viewed as a "soft-truncation" approach that pools contents dynamically instead of truncating the sequence brutally. To justify this compression process, we compared bucketing with truncation. For sequence truncation, we truncated the sequence from the left/right/middle side or a random position to fit the input into the summarization model. We also tested anchor-based hard truncation, which keeps only the top-30% anchors as input. Table 5 shows significant performance degradation when using hard-truncation, suggesting that it is more sensible to compress sequences dynamically according to importance than to truncate them brutally. However, cutting sequences based on anchor points still outperforms direct left/right/middle truncation. These results further demonstrate that anchors are informative tokens.

### 4.4 Computational Complexity

The reconstructor divides meetings of length $n$ into $r$ context-response pairs, where the average length of each context is $c$. The values of $n$, $r$, and $c$ are in the range of 5k-20k, 2-60, and 100-300, respectively. The time complexity of the reconstruction process is approximately $\mathcal{O}(r \times c^2 \times d_{model})$. For summarizer, our introduced RPB greatly com-

pressed the length of input from $n$ to $l$(1024 by default), without altering the model structure beyond its initial form. The time complexity is $\mathcal{O}(l^2 \times d_{model})$. Therefore, given the lengthy meeting texts, despite the additional introduction of the reconstructor, the combined complexity $\mathcal{O}(r \times c^2 \times d_{model}) + \mathcal{O}(l^2 \times d_{model})$ is much lower than that of the regular summary model, which has a complexity of $\mathcal{O}(n^2 \times d_{model})$. Our proposed approach effectively handles lengthy meeting texts with lower time complexity, making it a promising solution for real-world applications.

## 5 Related Work

Long-sequence processing techniques such as sliding-window attention (Beltagy et al., 2020), sparse sinkhorn attention (Tay et al., 2020), and hierarchical learning (Zhu et al., 2020; Rohde et al., 2021) are well-explored. These approaches target specifically lengthy input but ignore capturing salient information. Sentence compression (Shang et al., 2018a) and coarse-to-fine generation (Zhang et al., 2022) are developed to tailor the input length. However, the error propagation in intermediate steps severely limits their performance.

Meanwhile, language models have gradually equipped with dialogue acts (Goo and Chen, 2018), discourse relationship (Feng et al., 2021a), coreference resolution (Liu et al., 2021), and topic-segmentation (Liu et al., 2019; Li et al., 2019; Feng et al., 2021b). Despite the modest advances, these methods require external annotating tools or expert-grade annotators to accomplish the task. Feng et al. (2021b) employ the DialogGPT (Zhang et al., 2020b) as an annotator to capture keywords and topics. Despite the performance, adopting the token-wise loss to label keywords needs to be considered more deeply.

Additionally, cross-domain (Zhu et al., 2020) and large-scale dialogue-specific pretraining (Zhong et al., 2022) are utilized. However, large-scale datasets, excessive optimization steps, and sufficient computational resources are necessities and luxuries.

## 6 Conclusion

In this paper, we proposed RbS, a meeting summarization framework that accurately captures salient contents from noisy and lengthy transcripts. RbS uses a two-step process to evaluate content importance and dynamically compress the text to gen-

erate summaries. We introduce RPB, an anchor-based dynamic compression algorithm that condenses the original text, making RbS faster and more memory-efficient than one-step approaches. Without resorting to expert-grade annotation tools or large-scale dialogue-related pretraining tasks, experimental results on AMI and ICSI datasets show that RbS outperforms various previous approaches and reaches state-of-the-art performance.

## Limitations

Our exploration of the summary algorithm focuses on the traditional summary model. However, it is worth noting that in the era of large language models(LLM), effective compression of the input of LLM is worth being explored. Our future work should investigate how to effectively compress the input of LLM to make it more efficient.

## Ethical Considerations

We use publicly released datasets to train/dev/test our models. Generally, these previous works have considered ethical issues when creating the datasets. For the datasets we used in this work, we manually checked some samples and did not find any obvious ethical concerns, such as violent or offensive content. Source code and the models will be released with instructions to support correct use.

## Acknowledgements

We would like to thanks the anonymous reviewers for their valuable and constructive comments. This work was supported in part by the Hong Kong Innovation and Technology Fund(ITF) PRP/079/22FX, InnoHK initiative, the Government of the Hong Kong special administrative regions(HKSAR) of China, Laboratory for AI-Powered Financial Technologies.

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

## A  Appendix

### A.1  Training Details

The parameters of the models are initialized from Huggingface Libraries (Wolf et al., 2020) and updated by AdamW optimizer (Loshchilov and Hutter, 2019). All experiments are conducted on 8 A100 GPUs. The reconstructor and summarizer are both trained on the AMI and ICSI datasets. For the reconstruction, it takes 9 and 11 minutes to finish the 2,300 and 1,500 steps of training on the two datasets, respectively. The batch size is 256 and the learning rate is 5e-5. For the summarizer, it takes 8 and 9 minutes to finish the 100 steps of training. We use a batch size of 64 and a learning rate of 3e-5. Average scores of 3 runs are reported.

### A.2  Bucketing Algorithm

The size of the assigned buckets varies with the distance between the token and the anchor. The closer to the anchor tokens, the smaller the assigned buckets are. For the embedding of anchor tokens, the size of the assigned buckets is always 1. Given a maximum distance $d$, all token embeddings outside the anchors with distance $d$ will be uniformly assigned to the same bucket. The serial version of the bucketing algorithm is shown in the Algorithm 1.

### A.3  Re-using the Reconstructor as Summarizer

We conducted an ablation study to investigate the effect of initializing the summarizer with the weight of the reconstructor. The results showed that on AMI, changing the backbone led to a 0.99 ROUGE-1 score drop for BART-large and 1.11 for BART-Large-CNN. This suggests that with limited samples, directly migrating between response generation and meeting summarization introduces biases.

### A.4  Case Study

Table 6 compares the summarization generated by our framework and Summ^N. The red and cyan texts are the contents that appear in the gold summarization from AMI (Mccowan et al., 2005). In contrast, the magenta texts are the contents that are not in the gold summarization. orange texts in the gold summarization are the contents covered by RbS but ignored by the Summ^N. The violet texts are the common contents obtained by both RbS and Summ^N. RbS almost highlights all the key-points

**Algorithm 1:** Relative Positional Bucketing

**Input:** Relative Positions $\{R_1 \cdots R_n\}$
Bucket number $b$
Max distance $d$

1   $\{B_1, B_2, \cdots, B_n\} \leftarrow \emptyset$
2   $b \leftarrow b \mid 2$
3   **for** $i \leftarrow 1$ *to* $n$ **do**
4     **if** $R_i > 0$ **then**
5       $B_i \leftarrow B_i + R_i * b$
6     $R_i \leftarrow |R_i|$
7   **end**
8   $v \leftarrow b \mid 2$
9   **for** $i \leftarrow 1$ *to* $n$ **do**
10    **if** $R_i < v$ **then**
11     $B_i \leftarrow B_i + R_i$
12    **else**
13     $s \leftarrow v + \frac{\log(\frac{R_i}{v})}{\log(d/v)} * (b - v)$
14     $s \leftarrow \min(s, b - 1)$
15     $B_i \leftarrow B_i + s$
16    **end**
17 **end**

**Output:** Bucket Number$\{B_1, B_2, \cdots, B_n\}$

in the meeting, while Summ^N missed almost all the key points.

### A.5 More Visualization Examples

Figure 6 provide more visual samples of the heatmap to show more intuitively the anchor tokens selected by our framework. Keywords or phrases such as "selling price", "wholesale," and "retail" are captured by RbS compared with the other two different methods.

| Gold |
|------|
| The meeting opens with the group doing introductions by giving their name and role, betty is the project manager, francina is the user interface specialist eileen is the marketing expert and jeanne is the industrial designer. The project manager tells them they will be designing a new remote control that should be original trendy and userfriendly. They will be concerned with functional conceptional and detailed design. To try out the whiteboard, each group member draws their favorite animal on the board. They discuss the project budget and then talk about their experiences with remote controls. They seemed to agree that the remote should be compact and have a multi-purpose functions. They also agree that it should do something different that current controls cannot do, and that it should be made of different colors, materials, and shapes. They also discuss a way of helping people find the remote when it is lost, a signal whether it is a beep or light. Then they close the meeting with the project manager going over the tasks they are to complete and telling them they will meet again in about thirty minutes. Selling price will be twenty five euro. Company aims to profit fifty million euro. It should be compact, multi-functional, different in shape, color, material. Have a locator to help find the remote when it is lost. The industrial designer will work on the working design and technical function. The interface specialist will do the working design and functional design. The marketing manager will look for user requirement specifications, such as friendliness. The group is not sure if they will have the budget to make the gadget multi-functional, but they would like to make one that would control basically all household machines. |

| RbS |
|------|
| The project manager introduced the upcoming project to the team members and introduced the name and role of each participant in the project. The team then began a training exercise in which they learned how to use the white board, and practiced drawing on the whiteboard. The project manager also introduced the project budget and the projected profit aim of the project which was fifty million euros. The team then discussed their experiences with remote controls and what features they would like to see in the remote they will be producing. They discussed the features they would like to include in the remote control design, such as color options and different shapes. They also discussed the possibility of adding a locator function to help locate the remote when it is lost. They then discussed what features the remote should have and what price point it should be. The industrial designer will work on the working design. The user interface designer, and the marketing expert will work together on the technical design. Whether to have a light on the remote to help find the remote if it is misplaced. |

| Summ^N |
|------|
| The project manager introduced the project to the team members and went over the agenda. The team members discussed the project budget and discussed the features of the remote. The remote will control televisions, computers, and other household appliances. The group decided that the remote should be small, compact, and have a fancy look and feel. The industrial designer and user interface specialist will work on the technical and functional design. The marketing expert will work with the marketing expert to figure out how to sell the product. The project manager closes the meeting and the project manager gives each team member their individual assignments. They will get instructions to work with and if they have any questions, they can ask them. It was decided that it would be a good idea to include a throw signal to help locate the remote when it is lost. It would be possible to make the remote more fashionable by using different colors and materials and using different shapes. The device will be for televisions only, and will not be for teletext. It should be a multi-functional gadget that controls all household machines. It could be used for voice recognition as well as voice recognition. It will be made of plastic and rubber and will be shaped like a kidney. There will be no LCD screen, and the remote will have buttons for power, volume, mute, channel-changing, channel up/down, channel down, and mute. They were not sure how much the remote would cost to produce. They did not know what the profit aim was for the project. They decided to use a white board to draw their favorite animals on the white board. They also decided to include an indicator on the remote so that it will light up when a button is pressed. They discussed how to incorporate the company colors and logo into the design. |

Table 6: Case study of RbS

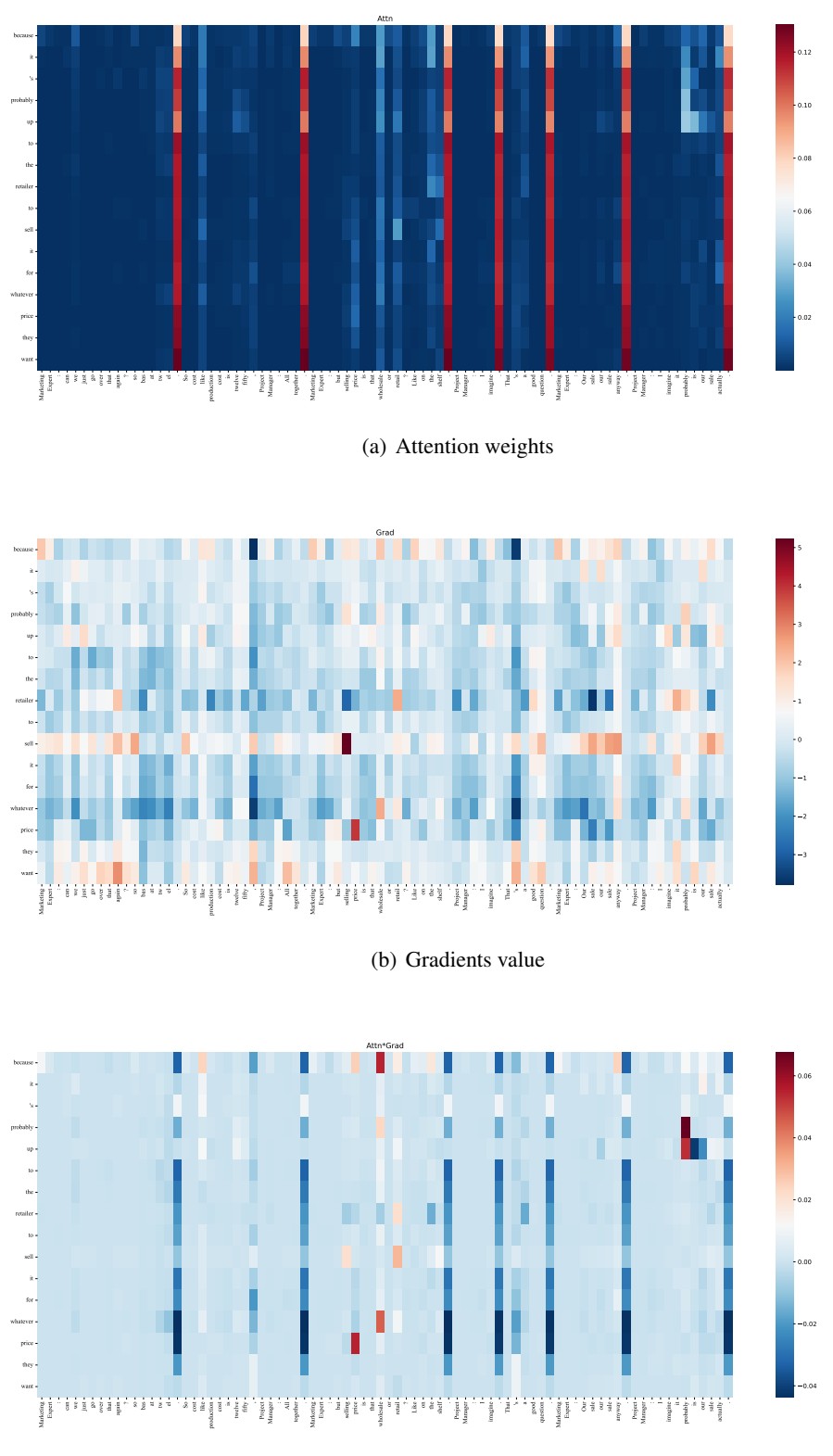

(a) Attention weights

(b) Gradients value

(c) Scaled attentions

Figure 6: Visualization of the attention map, gradients, and the scaled attention weights