# OpenReview forum: "Reconstruct Before Summarize: An Efficient Two-Step Framework for Condensing and Summarizing Meeting Transcripts"
_EMNLP/2023/Conference — EMNLP 2023 Main_

### Official Review · Reviewer_KpoB · 2023-08-04

**Typos Grammar Style And Presentation Improvements:** Paper text needs proof-reading/gramma…
**Soundness:** 4

**Excitement:**

4: Strong: This paper deepens the understanding of some phenomenon or lowers the barriers to an existing research direction.

**Paper Topic And Main Contributions:**

The paper proposes a two-step framework for the abstractive summarization of meetings, Reconstruct Before Summarize (RbS). It works by first, selecting the key points in the input (anchor tokens) in a self-supervised way, and second, summarizing using the Relative Position Bucketing algorithm which compresses the input and makes the method more memory and computationally efficient.

With BART-large/BART-large-CNN as the backbone, the authors evaluate their approach on AMI and ICSI datasets against a series of baselines and find out that RbS attains the best performance on both datasets and most metrics (Rouge-1/2/L). They further perform ablation study on the contribution of anchor tokens, scoring methods and bucketing to the overall method.

**Reasons To Accept:**

* A new method for long input summarization is proposed
* It attains state-of-the-art results among non-LLM methods in terms of Rouge score
* anchoring and bucketing also results in a reduced time complexity

**Reasons To Reject:**

* description of the methodology is mostly unclear
* "Reconstruct Before Summarize" is not fluent - this could have been in the typos/grammar section as well, but it's in the paper title

**Reproducibility:**

2: Would be hard pressed to reproduce the results. The contribution depends on data that are simply not available outside the author's institution or consortium; not enough details are provided.

**Reviewer Confidence:**

4: Quite sure. I tried to check the important points carefully. It's unlikely, though conceivable, that I missed something that should affect my ratings.

---

> ### Author Rebuttal · Authors · 2023-08-29
>
> Thank you for reviewing our submission and providing us with your valuable feedback. Your feedback has been immensely helpful in improving our work. We would like to address your concerns as follows:
>
> ***
> **Concern**: The description of the methodology is mostly unclear
>
> **Response**:
> Regarding the methodology, we apologize for any confusion caused and fully understand the importance of providing a clear and concise explanation. In the final version, we will provide a more detailed explanation of our methodology, including the bucketing and anchoring algorithms, in a manner that is easy to understand. We appreciate your suggestion and will make every effort to ensure that the methodology is presented in a professional manner.
> ***
>
> **Concern**: evaluation issue: automatic-only (furthermore, word overlap-only) metrics are not enough in assessing summarization accuracy
>
> **Response**:  With regard to the evaluation issue, we agree that automatic-only and word overlap-only metrics may not be sufficient in assessing summarization accuracy. As a result, we attempted to organize a human evaluation during the rebuttal period.
> For the generated summaries released in the DialogLM repo, they didn't indicate the corresponding transcript for each summary. Given one generated summary produced by DialogLM, we had to ask the reviewers to try to figure out its corresponding meeting transcript, and then evaluate the quality. The extraordinarily lengthy transcripts made their work extremely difficult. In the end, we performed a human evaluation of only five samples due to the workload. The results of the remaining samples will be finalized and added to the final version.
>
> The human evaluation involved two postgraduate MSc. students and an individual engaged in clerical work. They were asked to determine, based on the meeting transcripts, which generated summaries that better covered the important decisions, actions to be taken, and problems encountered in the meeting. They were also required to compare the quality of the final summaries generated. Throughout the process, they were not told which model the results came from. We aggregated the results from all three reviews and tallied the RbS's winning rate as below.
> ***
> |               | Win Rate |
> |---------------|----------|
> | Decisions     | 3 / 5    |
> | Actions       | 3 / 5    |
> | Problems      | 4 / 5    |
> | Final Quality | 4 / 5    |
> ***
> We observed that in each data sample, the final summary generated by our framework contained a summary of the meeting procedure, followed by the actions to be taken, the meeting decisions, and the problems encountered in the meeting. In contrast, many important decisions, actions, and problems were not covered in the summaries generated by DialogLM. We then found that for all DialogLM's official released generated summaries in their github repo, **twelve out of twenty AMI summaries** are incomplete results. We speculate that the reason behind this is that after DialogLM generates a part of the summary, extremely long inputs and outputs consume a lot of memory, making it impossible to generate more tokens due to memory limitations. This phenomenon was also observed in the ICSI dataset, where **three of the six results** of DialogLM were extremely incomplete.
>
> ***
> We appreciate your comments and will make every effort to fix the grammatical errors in the final version and to present the methodology in a more understandable manner. Thank you for your help and for the opportunity to address your concerns.

---

### Official Review · Reviewer_9DKz · 2023-08-12

**Typos Grammar Style And Presentation Improvements:** Reconstrcut before Summarize(RbS)  li…
**Soundness:** 4

**Excitement:**

3: Ambivalent: It has merits (e.g., it reports state-of-the-art results, the idea is nice), but there are key weaknesses (e.g., it describes incremental work), and it can significantly benefit from another round of revision. However, I won't object to accepting it if my co-reviewers champion it.

**Missing References:**

none

**Paper Topic And Main Contributions:**

The paper proposes a two-step approach to summarizing long meeting transcripts called Reconstruct before Summarize (RbS). The first step includes reconstruction via the forward pass of language model with a teacher forcing and then finding silent words by retracing the anchor tokens. The contribution scores of the anchor tokens candidates are either averaged or counted with multi-view voting). Then the context window surrounding the anchors is compressed and embedded for the summarizer based on BART or BART-CNN. The originally long transcript is first compressed (by means of relative positional bucketing) to its salient essence and then subject to summarization, allowing for significant size reduction while still preserving the most critical information (i.e. in the AMI dataset from the average ca 6000 to 300 words). The authors claim their concept to be effective (outperforming other models in ROUGE-1, ROUGE-2, and ROUGE-L metrics on AMI and ICSI datasets) and particularly efficient since it does not require large language models or additional annotation of salient words.

**Questions For The Authors:**

None

**Reasons To Accept:**

In the era of solving every task with large language models, it is especially noteworthy when some effort is made to enhance the old-fashioned methods based on less costly and energy-consuming models. Moreover, since the proposed method is effective (as shown in the results) and based on a deeper understanding of the data and the task, it is a clear advantage. Some hypotheses were made about the particular text structure resulting from transcribing a multi-speaker meeting, like the distance of salient information from an anchor token, and based on them, a novel variation of existing approaches is introduced. The paper is mainly well-written; the explanations should allow for replicas. Other approaches has been briefly discussed and explained. The visualizations and tables are clear and helpful. Beside the main report on experimental setup and results there is also a deeper analysis offered.

**Reasons To Reject:**

One of the central claims is the model's efficiency - but in the appendix, the reported computation time includes 8 A100 GPUs which is a luxury itself. Reporting the computation time on a more moderate set would be only fair.
Especially, since the conlusion reads "However, large-scale datasets, excessive optimization steps, and sufficient computational resources are necessities and luxuries".
The limitation section only includes focus on LLMs, whereas there are other directions to be explored, such a comparison with a linguistic-based (based on the domain knowledge of e.g. speech acts) approach could be a good compliment to the study.


**Reproducibility:**

4: Could mostly reproduce the results, but there may be some variation because of sample variance or minor variations in their interpretation of the protocol or method.

**Reviewer Confidence:**

4: Quite sure. I tried to check the important points carefully. It's unlikely, though conceivable, that I missed something that should affect my ratings.

---

> ### Author Rebuttal · Authors · 2023-08-28
>
> Thank you for your feedback on our paper. We appreciate the time and effort you put into reviewing our work. We have carefully considered your comments and suggestions and have made the following responses:
> ***
> **Concern**: 8 A100 GPUs are luxuries.
>
> **Response**: Indeed, 8* A100 are luxries. However, it is worth noting that our approach is platform-independent. That is, no matter what platform the algorithm runs on, our proposed RbS is more efficient than most baselines(Section 4.4). We use 8 * A100 not because our algorithm relies on the efficiency of the A100 platform, but because we have no other choice but to use A100.
>
> One important issue is that when truncating the meeting transcripts to 3072 - 5120 for the training of BART baselines, the batch size can only be set to less than 4 * 8 = 32 on 8 40G A100, and the batch size is even lower when untying the weights of the embedding and the output layer. In our approach, the total memory usage is only 71.18% with a total batch size of 64, we didn't further increase the batch size because the size of the training set is too small(97 samples for AMI, 42 samples for ICSI). This indicates that the advantages of our approach in terms of memory usage will be even more pronounced on lower-resource computing platforms.
>
> Besides, It is worth noting that DialogLM is also trained and tested on 8 * A100 platforms. However despite this abundance of computational resources, many important decisions, actions, and problems were not covered in the summaries generated by DialogLM. We found that for all the DialogLM's official released generated summaries in their github repo, **twelve out of twenty AMI summaries** are incomplete texts. We speculate that the reason behind this is that after DialogLM generates a part of the summary, extremely long inputs and outputs consume a lot of memory, making it impossible to generate more tokens due to memory limitations. This phenomenon was also observed in the ICSI dataset, where **three of the six summaries** of DialogLM were extremely incomplete and unreadable.
>
> Compared to DialogLM, we have the same 8*A100 computational platform, but our summaries are complete and are not constrained by the memory pressure imposed by sequence length. Our results cover the decisions, actions, and problems encountered in the meeting transcripts.
> ***
>
> **Concern**: There are other directions to be explored, such as a comparison with a linguistic-based (based on the domain knowledge of e.g. speech acts) approach could be a good compliment to the study.
>
> **Response**: We truly appreciate that you pointed this out, we think it will help us improve the overall completeness of the paper. We compared some of the approaches that utilize linguistic knowledge to generate abstracts, such as dialogue acts[1], and discourse relationships[2]. Indeed, we emphasized in the paper that we annotate the key points without the help of specialized linguistic tools. However, we did not analyze them in depth in the paper. I apologize for this, and we will do a more in-depth comparison and analysis in the final version.
> ***
>
> Thank you again for your thoughtful feedback!
>
> ## References
> [1] [Abstractive Dialogue Summarization with Sentence-Gated Modeling Optimized by Dialogue Acts](https://ieeexplore.ieee.org/document/8639531)(Goo et al., IEEE SLT 2018)
>
> [2] [Dialogue Discourse-Aware Graph Model and Data Augmentation for Meeting Summarization](https://www.ijcai.org/proceedings/2021/0524) (Feng et al., IJCAI 2021)

---

### Official Review · Reviewer_NpwE · 2023-08-20

**Soundness:** 4

**Excitement:**

3: Ambivalent: It has merits (e.g., it reports state-of-the-art results, the idea is nice), but there are key weaknesses (e.g., it describes incremental work), and it can significantly benefit from another round of revision. However, I won't object to accepting it if my co-reviewers champion it.

**Paper Topic And Main Contributions:**

This paper proposes a new algorithm for summarizing meeting minutes. Motivated by the fact that meeting transcripts tend to be long and that the topics vary within a single meeting, authors propose a two-step approach to obtain meeting summaries. The first stage is responsible for finding the pivotal keywords and expressions according to the "reconstructor" model, by utilizing the scaled attention weights by the model. To do this, BART is fine-tuned on the response generation task. After annotating the important words in the meeting, the second stage groups tokens centered around each important keyword and aggregate them according to relative positions. Finally, the aggregated embeddings are used to generate the summary.

Authors experimented the proposed method on AMI and ICSI and obtained state-of-the-art results in terms of ROUGE scores. Analyses were conducted for each stage to confirm the role and positive impacts of each proposal toward the final results.

**Questions For The Authors:**

* For the ablation analysis in 4.1, naively deleting or substituting the anchor tokens would corrupt the syntactic and semantic structure, not to mention that it’s done at  token-level. Have authors accounted for this?
* For the same analysis, does "high-frequency tokens" imply that many of them are function words? Or were those excluded?

**Reasons To Accept:**

* Proposed mechanisms are reasonable and well-motivated, leading to improvement in ROUGE scores.
* Extensive ablation study is conducted to support the validity of each component.
* Outcome is the state-of-the-art method for both AMI and ICSI.

**Reasons To Reject:**

* Some key details are missing, making it difficult to understand how exactly the summaries are generated. In particular, the explanation on the summarizer is lacking (after aggregating the embeddings, how to get the final summary).
* Besides the time-complexity analysis, it would have been nice to run and compare the efficiency between the proposed method and baselines.
* While the results look better, it’d be better to run statistical testing to support statistical significance.

**Reproducibility:**

3: Could reproduce the results with some difficulty. The settings of parameters are underspecified or subjectively determined; the training/evaluation data are not widely available.

**Reviewer Confidence:**

4: Quite sure. I tried to check the important points carefully. It's unlikely, though conceivable, that I missed something that should affect my ratings.

**Typos Grammar Style And Presentation Improvements:**

* Add spaces before parentheses in the Figure 1 caption.
* L277 "DialogGPT" instead of DialoGPT
* L214 remove "the" T5

* Figure 5: Too small to read. Use smaller figure size dimensions for better readability

---

> ### Author Rebuttal · Authors · 2023-08-27
>
> Thank you for taking the time to review our paper and for providing valuable feedback. We appreciate your thoughtful comments and suggestions, which help us to improve the quality of our work.
> ***
> **Concern**: After aggregating the embeddings, how to get the final summary?
>
> **Response**: Regarding your concern about how our summarizer generates the final summary after aggregating the embedding, we apologize for any confusion caused by our writing. We have made revisions to our paper to clarify the process.
>
> The difference between our summarizer and the original BART is only the addition of the relative postitional bucketing(RPB) module, which is inserted between the embedding layer and the first attention layer. RPB compresses the original long embedding to an acceptable length, the rest of the process remains the same. As usual, BART takes the compressed embeddings and processes them with MLP and attention layers, and generates the final summarization. The whole process is not different from the original BART, except that BART gets the compressed embedding instead of the original one.
> ***
>
> **Concern**: Run and compare the efficiency between the proposed method and baselines.
>
> **Response**: We also appreciate your suggestion to compare the efficiency of our proposed method with baselines. We will include a comparison of runtimes and memory consumption in the final version of our paper. However, one important issue is that when truncating the meeting transcripts to 3072 - 5120 for the training of BART baselines, the batch size can only be set to less than 4 * 8 = 32 on 8 40G A100, and the batch size is even lower when untying the weights of the embedding and the output layer. In our approach, the total memory usage is only 71.18% with a total batch size of 64, we didn't further increase the batch size because the size of the training set is too small(97 samples for AMI, 42 samples for ICSI). RbS can training with a larger batch size, which not only improves learning efficiency, but also reduces I/O time. We will include a detailed comparison of computation time and I/O time in the final version.
> ***
> **Concern**: Statistical testing
>
> **Response**: We have added a comparison of statistical testing relative to DialogLM and Summ^N, as you suggested.
> It is worth noting that, since we cannot reproduce their results, we chose their results in the original paper as a reference, and we take their ***average results of three runs*** in the original paper and calculate the p-values. The results reported in the DialogLM paper are the average of three runs, while the Summ^N results do not specify whether their results are the average or the maximum. Here we default the Summ^N results to the average of three runs as well. The results of the significance tests are as follows:
> 1. On the AMI:
>     - RbS vs. DialogLM: The z-value is 1. The p-value is .15866. The result is **not significant** at **p < 0.10**.
>     - RbS vs. Summ^N: The z-value is 2.33333. The p-value is .00982. The result is **significant** at **p < 0.01**.
>     - RbS-CNN vs. DialogLM: The z-value is 3. The p-value is .00135. The result is **significant** at **p < 0.01**.
>     - RbS-CNN vs. Summ^N: The z-value is 3. The p-value is .00135. The result is **significant** at **p < 0.01**.
>
> 2. On the ICSI:
>     - RbS vs. DialogLM: The z-value is 2.33333. The p-value is .00982. The result is **significant** at **p < 0.01**.
>     - RbS vs. Summ^N: The z-value is 3. The p-value is .00135. The result is **significant** at **p < 0.01**.
>     - RbS-CNN vs. DialogLM: The z-value is 1. The p-value is .15866. The result is **not significant** at **p < 0.10**.
>     - RbS-CNN vs. Summ^N: The z-value is 3. The p-value is .00135. The result is **significant** at **p < 0.01**.
> ***
> **Concern**: naively deleting or substituting the anchor tokens would corrupt the syntactic and semantic structure
>
> **Response**: We partially agree with this point of view. However, it is noticed that for meeting transcripts with 5k+ tokens, with 6.4% anchor tokens, removing 25% anchors normally only deletes around 80 tokens. Usually, removing/replacing 80 words in a sequence of 5000 words, from an intuitive point of view, will not have a great impact on the semantics of the original sequence. Unless extremely important keywords are removed. So our experiment proved to a certain extent that the anchor words are mostly some keywords.
>
> Besides, to further prove the effectiveness of the anchors, we did an experiment in the ablation study(See Table 2.), that is, sorting all the anchors according to the importance score and taking 25% as a fraction, and then removing one fraction at a time to see the changes of the performance. It can be seen that removing the top 25% of anchors causes the greatest loss of information in the original text. Through this experiment, we proved that anchors with higher importance scores are more important.
> ***
> **Concern**: Does "high-frequency tokens" imply that many of them are function words? Or were those excluded?
>
> **Response**:
> We apologize for overlooking your point about whether "high-frequency tokens" imply that many of them are function words or not. For some transcripts, we found that the high-frequency words are mainly filler words and stop words, while for others, high-frequency words are mainly some high-frequency nouns in the meeting content. In our final version, we will consider replacing the anchors with high-frequency words other than filler words and stop words to further verify the effectiveness.
> ***
>
> We are very grateful for your feedback regarding the typos, grammar errors, and readability of the figures. We will make the necessary corrections in the final version.
> Thank you again for your valuable feedback.

---

### Official Review · Reviewer_hWuH · 2023-08-22

**Soundness:** 3

**Excitement:**

3: Ambivalent: It has merits (e.g., it reports state-of-the-art results, the idea is nice), but there are key weaknesses (e.g., it describes incremental work), and it can significantly benefit from another round of revision. However, I won't object to accepting it if my co-reviewers champion it.

**Paper Topic And Main Contributions:**

This paper proposes the approach, Reconstruct Before Summarize, for meeting summarization, where the contexts are lengthy and noisy casual conversations.
RbS consists of a reconstructor and a summarizer. The reconstructor locates the salient information when reconstructing the next utterance. The summarizer compresses the long context based on the distance to the salient information and generates a summary. The performance achieves SOTA on meeting summarization.

**Reasons To Accept:**

1.  It is nice to leverage dialogue history as an unsupervised objective in the stage of reconstruction. The tokens in previous utterances that have a greater impact on the next utterance are more likely to contain salient information. This method is reasonable and effective, as shown in Fig 4.
2. Another highlight of this article is the compression of redundant meeting transcripts. The anchor token selection and Relative Positional Bucketing provide a simple but effective solution in this situation.

**Reasons To Reject:**

1.	The datasets for evaluation are both in a limited scale. AMI has 20 test samples and ICSI has 6. This hurts the confidence of the generalization. Any data augmentation method is performed? Or the author may consider other evaluations to prove the generalization for scalability.
2.	The related work review provides little information on the task of meeting summarization. Also, in Table 1, the compared baselines are not designed for the task of meeting summarization. In line 516, the complexity of regular summary models is  O(n2 × d_model). Does this mean that some previous method inputs the long meeting scripts into a model?
The lacking of background makes the contributions of this paper ambiguous.

**Reproducibility:**

4: Could mostly reproduce the results, but there may be some variation because of sample variance or minor variations in their interpretation of the protocol or method.

**Reviewer Confidence:**

3: Pretty sure, but there's a chance I missed something. Although I have a good feel for this area in general, I did not carefully check the paper's details, e.g., the math, experimental design, or novelty.

---

> ### Author Rebuttal · Authors · 2023-08-27
>
> Thank you for taking the time to review our paper on meeting summarization. We appreciate your feedback and have carefully considered all of your concerns.
> ***
> **Concern**: Any data augmentation method is performed?
>
> **Response**: Regarding your first concern about data augmentation methods, we did not use any such methods in our framework. While we acknowledge that successful data augmentation approaches could further improve results, we intend to isolate and highlight the effectiveness of our proposed method.
> ***
> **Concern**: AMI has 20 test samples and ICSI has 6, this hurts the confidence of the generalization.
>
> **Response**: We understand your concern about the relatively small amount of data tested in our study, with AMI having 20 test samples and ICSI having 6. However, it is worth noting that multiple previous works[1-4] in the field of meeting summarization have validated their results on these datasets.
> Additionally, these datasets are regarded as the definitive benchmarks in surveys[5, 6] related to meeting summarization due to their extremely realistic meeting scenarios and accurate labeling of decisions, actions, problems, and meeting minutes. Despite the limited amount of training data and the long sequence lengths, these datasets remain significant challenges in the field.
> ***
> **Concern**: the compared baselines are not designed for the task of meeting summarization
>
> **Response**: Regarding your concern about the compared baselines not being designed for meeting summarization, we respectfully disagree. In Table 1, with the exception of our backbone BART model, all other methods[1-4, 7] are standard baselines in the meeting summarization field. These methods have been heavily experimented with on AMI, and those capable of dealing with extra long sequences have been tested on ICSI. In surveys[5, 6] related to meeting summarization, these methods are regarded as landmark methods in the field.
> ***
> **Concern**: Does this mean that some previous method inputs the long meeting scripts into a model? The lacking of background makes the contributions of this paper ambiguous.
>
> **Response**: We truly appreciate your feedback on the lacking background of our contributions. In meeting summarization, the majority of work chooses to truncate original texts, with the truncated text typically between 3072 and 8192 in length depending on computational resources. Even after sacrificing raw information to save memory overhead and time, the time complexity of truncating a very long sequence to 3072 is still larger than our framework, and the accuracy lags far behind ours(HAT-BART[8]). Meanwhile, expanding the sequence to 5120[4] or 8192[9] consumes more memory. Our plug-and-play RPB module only applies pooling operation on the embeddings, which greatly enhances generalizability compared to frameworks that rely on sparse attention or longformer-based techniques.
>
> It is also worth noting that during the inference, baselines that use more memory to retain the original lengthy input will have less memory reserved for the output. As in DialogLM's official released AMI and ICSI generation results, more than **50%** of the summaries are incomplete texts. Whereas we are not limited by sequence length, our proposed RbS can efficiently and accurately cover all decisions, actions, and problems in the meeting transcripts.
> ***
> Thank you for your valuable feedback. We will take your suggestions into account and improve our final version accordingly.
>
> ## References
> [1] [Keep Meeting Summaries on Topic: Abstractive Multi-Modal Meeting Summarization](https://aclanthology.org/P19-1210) (Li et al., ACL 2019)
>
> [2] [A Hierarchical Network for Abstractive Meeting Summarization with Cross-Domain Pretraining](https://aclanthology.org/2020.findings-emnlp.19) (Zhu et al., Findings 2020)
>
> [3] [SummN: A Multi-Stage Summarization Framework for Long Input Dialogues and Documents: A Multi-Stage Summarization Framework for Long Input Dialogues and Documents](https://aclanthology.org/2022.acl-long.112) (Zhang et al., ACL 2022)
>
> [4] [Dialoglm: Pre-trained model for long dialogue understanding and summarization](https://ojs.aaai.org/index.php/AAAI/article/view/21432/21181) (Zhong et al., AAAI 2022)
>
> [5] [Meeting Summarization: A Survey of the State of the Art](https://arxiv.org/pdf/2212.08206) (Kumar, L. P., & Kabiri, A. Arxiv 2022)
>
> [6] [Abstractive meeting summarization: A survey](https://doi.org/10.1162/tacl_a_00578) (Rennard et al., Transactions of the Association for Computational Linguistics (2023) 11: 861–884.)
>
> [7] [Dialogue Discourse-Aware Graph Model and Data Augmentation for Meeting Summarization](https://www.ijcai.org/proceedings/2021/0524) (Feng et al., IJCAI 2021)
>
> [8] [Hierarchical Learning for Generation with Long Source Sequences](https://arxiv.org/abs/2104.07545) (Rohde et al., Arxiv 2021)
>
> [9] [ConvoSumm: Conversation Summarization Benchmark and Improved Abstractive Summarization with Argument Mining](https://aclanthology.org/2021.acl-long.535) (Fabbri et al., ACL-IJCNLP 2021)

---

### Meta-Review · Area_Chair_855n · 2023-09-19

**Recommendation:** 5

**Metareview:**

The paper brings merits of a reconstructor locates the salient information when reconstructing the next utterance, and improves meeting summarization. The paper is expected to added more details and discussions as mentioned in the rebuttal.

---

### Decision · Program_Chairs · 2023-10-07

**Decision:**

Accept-Main

**Comment:**

The paper brings merits of a reconstructor locates the salient information when reconstructing the next utterance, and improves meeting summarization. The paper is expected to added more details and discussions as mentioned in the rebuttal.